# COVID-19 Infection during Pregnancy: Disruptions in Lipid Metabolism and Implications for Newborn Health

**DOI:** 10.3390/ijms241813787

**Published:** 2023-09-07

**Authors:** Natalia Frankevich, Alisa Tokareva, Vitaly Chagovets, Natalia Starodubtseva, Natalia Dolgushina, Roman Shmakov, Gennady Sukhikh, Vladimir Frankevich

**Affiliations:** 1National Medical Research Center for Obstetrics Gynecology and Perinatology Named after Academician V.I. Kulakov of the Ministry of Healthcare of Russian Federation, 117997 Moscow, Russia; n_frankevich@oparina4.ru (N.F.); a_tokareva@oparina4.ru (A.T.); v_chagovets@oparina4.ru (V.C.); n_starodubtseva@oparina4.ru (N.S.); n_dolgushina@oparina4.ru (N.D.); r_shmakov@oparina4.ru (R.S.); g_sukhikh@oparina4.ru (G.S.); 2Moscow Institute of Physics and Technology, 141700 Moscow, Russia; 3Department of Obstetrics, Gynecology, Perinatology and Reproductology, Institute of Professional Education, Federal State Autonomous Educational Institution of Higher Education, I.M. Sechenov First Moscow State Medical University, Ministry of Health of the Russian Federation (Sechenov University), 119991 Moscow, Russia; 4Laboratory of Translational Medicine, Siberian State Medical University, 634050 Tomsk, Russia

**Keywords:** SARS-CoV-2, COVID-19, pregnancy, plasma, metabolomics, lipidomics, mass spectrometry, biomarkers

## Abstract

The COVID-19 pandemic has raised questions about indirect impact in pregnant women on the development of their future children. Investigating the characteristics of lipid metabolism in the “mother–placenta–fetus” system can give information about the pathophysiology of COVID-19 infection during pregnancy. A total of 234 women were included in study. Maternal plasma, cord blood, and amniotic fluid lipidome were analyzed using HPLC-MS/MS. Differences in lipid profile were searched by Mann–Whitney and Kruskall–Wallis test, and diagnostic model based on logistic regression were built by AIC. Elevated levels of lysophospholipids, triglycerides, sphingomyelins, and oxidized lipids were registered in patients’ maternal and cord plasma after COVID-19 infection. An increase in maternal plasma sphingomyelins and oxidized lipids was observed in cases of infection during the second trimester. In amniotic fluid, compared to the control group, nine lipids were reduced and six were elevated. Levels of phosphoglycerides, lysophosphoglycerides, and phosphatidylinositols decreased during infection in the second and third trimesters of pregnancy. A health diagnostic model for newborns based on maternal plasma was developed for each group and exhibited good diagnostic value (AUC > 0.85). Maternal and cord plasma’s lipidome changes during delivery, which are associated with COVID-19 infection during pregnancy, are synergistic. The most significant disturbances occur with infections in the second trimester of pregnancy.

## 1. Introduction

Respiratory viral infections are the most frequently reported infections during pregnancy. It is generally accepted that they do not cross the placental barrier and are therefore considered not dangerous for pregnant women [1]. The COVID-19 pandemic raged from 2019 to 2023; its end has raised new questions about the possible indirect impact of respiratory infections in pregnant women on the proper development of their children in the future [2].

A considerable amount of data collected to date indicates that pregnant women have a similar susceptibility to infection with SARS-CoV-2, the virus that causes COVID-19, and a resulting severe illness as other young people. However, there are sporadic data on cases of vertical transmission of the infection [3,4,5].

The rate of preterm birth, i.e., birth before 37 weeks of gestation, is 10% (WHO, 2018); this is the same as the rate of preterm birth in pregnant women with COVID-19. Analysis of data on epidemics caused by SARS and MERS coronaviruses did not show a correlation with an increase in the incidence of severe malformations [6].

Further research is needed to understand the accumulated material and to gain a deep understanding of the pathogenesis and epidemiology of SARS-CoV-2 during pregnancy, including aspects such as the gestational age at which the mother was infected, the severity of the infection, the impact of concomitant diseases, and the frequency of adverse outcomes. These data can be applied to new epidemics of respiratory viral infections. At the moment, there is no objective data on severe forms of COVID-19 and obstetric complications in early pregnancy, and there is not enough information about the impact of the infection on the course of pregnancy, its outcomes, or the newborn at different stages.

Using HPLC-MS/MS analysis to identify metabolites in the mother–placenta–fetus system, particularly lipidome, can offer new insights into the pathophysiology of severe respiratory viral diseases during pregnancy and potentially serve as predictors of adverse pregnancy outcomes. Lipids play a crucial role in fetal growth and development, and changes in their concentration in the mother’s body during the second and third trimesters are directly related to the fetus’s maximum growth during this period. A study by Dionne V. Gootjes et al. explored the relationship between the mother’s lipid profile in early pregnancy and the size of the embryo, revealing an association between increased triglyceride and residual cholesterol concentrations in early pregnancy with embryo size [7]. Nevertheless, the gestational age at which these lipid concentrations critically affect the intrauterine development of the fetus remains uncertain.

Limited knowledge exists on how viral infection exposure and severity affect metabolism, including the lipid profile in the mother–placenta–fetus system. Prenatal risk factors can have both immediate and long-term effects on human health [8,9]. The possibility of active virus transmission from mother to fetus or newborn during pregnancy and childbirth remains a subject of debate [5]. Despite this uncertainty, newborns of mothers infected with COVID-19 may be at a high risk of complications in the early neonatal period and may face long-term health consequences.

In light of these concerns, the aim of this study was to investigate the characteristics of the lipid profile in the mother–placenta–fetus system among women who had experienced SARS-CoV-2 infection during pregnancy.

## 2. Results

### 2.1. Features of Pregnancy Course and Outcomes with COVID-19 Infection

During the period from September 2020 to February 2022, a total of 234 mother–newborn couples were prospectively enrolled in the case-control study to investigate the impact of COVID-19. Based on the presence or absence of COVID-19 during pregnancy, the patients were categorized into two groups. Group I (main) consisted of patients who had COVID-19 during pregnancy (*n* = 175). This group was further stratified into three subgroups: subgroup 1a—patients with COVID-19 in the first trimester (*n* = 41), subgroup 1b—patients with COVID-19 in the second trimester (*n* = 67), and subgroup 1c—patients with COVID-19 in the third trimester (*n* = 67). Group II (control) included patients who did not have COVID-19 during pregnancy (*n* = 59) (Appendix A).

The analysis aimed to identify factors associated with the severity of the new coronavirus infection in 2019 and the frequency of complications for both the mother and fetus. Comprehensive assessments were conducted on anthropometric data, somatic and gynecological history, concurrent somatic conditions, reproductive function and parity, pregnancy course, postpartum period, and the newborn’s condition at birth and during hospital stay (Table 1 and Table 2). However, no statistically significant differences were observed between the groups (*p* > 0.05).

In 14.6% of cases, puerperas who experienced COVID-19 in the first trimester had a higher incidence of preterm labor (*p* = 0.03). Additionally, newborns from this group received lower Apgar scores at 5 min (*p* = 0.009), and only 30% of cases were classified as “healthy” (*p* < 0.001) (Table 3).

Among pregnant women who contracted COVID-19 in the second trimester, 19.4% experienced gestational diabetes mellitus (GDM) without the need for insulin therapy (*p* = 0.01). Preterm labor occurred in 11.9% of cases in this group (*p* = 0.03), and the frequency of emergency cesarean sections increased to 18.2% (*p* = 0.01). Moreover, only 43.3% of newborns from this group were classified as “healthy” (*p* = 0.01). Notably, a higher percentage of newborns in this group had minor heart anomalies (33.4%, *p* = 0.02), and 19.7% showed the presence of an additional chord among CHD (*p* < 0.001). In cases of COVID-19 in the third trimester, only 38.8% of newborns were assigned to the “healthy” group (*p* = 0.002).

### 2.2. COVID-19 Related Changes in Maternal Plasma, Cord Blood and Amniotic Fluid Lipidome

A semi-quantitative analysis of lipidomics was conducted on three biological fluids (maternal blood plasma, cord blood plasma, and amniotic fluid) in 234 patients during the delivery stage. The positive ion mode identified 180 lipids in maternal plasma, 137 lipids in cord plasma, and 192 lipids in amniotic fluid. In the negative ion mode, 186 lipids were identified in maternal plasma, 159 in cord plasma, and 209 in amniotic fluid. These lipids belong to various classes, including cholesterol esters (CE), diglycerides (DG), ceramides (Cer), cardiolipins (CL), (lyso)phosphatidylcholines (LPC, PC), (lyso)phosphatidylethanolamines (LPE, PE), monogalactosyldiacylglycerides (MGDG), oxidized lipids (OxL-), plasmanyl- and plasmenyl-phospholipids (Plasmanyl- and Plasmenyl-), phosphatidylinositols (PI), (lyso)phosphatidylglycerols (LPG, PG), dimethylphosphatidylethanolamines (DMPE), sphingomyelins (SM), and triacylglycerides (TG) (Appendix A).

Significant differences were observed between the group of healthy patients and those who had contracted COVID-19 in terms of several lipids in maternal plasma (Figure 1, Appendix A). Within the maternal plasma of the primary group, a discernible increase was observed in OxLPC18:3(OOO), OxTG 16:0_18:1_16:1(CHO), OxTG 18:0_18:1_16:1(COOH), OxPC 18:0_18:4(OO), OxPC 20:4_16:1(COOH); conversely, there was a decrease in OxCL 18:1(OH)_18:1(OH)_22:6_22:6 and OxPC 18:2_14:1(COOH). Notably, an extensive array of lipids that exhibited statistically significant alterations during the COVID-19 period (LPC, TG, PC, and CL) were indeed found to be oxidized. Significantly, it is primarily oleic acid C18:1 and palmitoleic acid C16:1 that undergo notable oxidation in this context. The levels of lysophospholipids and triglycerides increased, with the increase being more pronounced if the infection occurred later in pregnancy. Infections in the second trimester were associated with a marked increase in maternal plasma sphingomyelins and oxidized lipids (CL, LPC, PC, PE) (Appendix A). However, plasma PI levels decreased with a history of COVID-19.

The impact of COVID-19 infection on the lipid composition of cord blood plasma was found to be significant (Figure 2, Appendix A). In the umbilical cord plasma of the primary group, there was a notable rise observed in OxCL 18:1_18:3(OOH)2_18:3(OOH)2_20:3, OxPC 16:0_20:3(1O), OxPC 20:2_16:1(COOH), OxPC 22:4_16:1(Ke, OH), OxPC 22:6_16:1(COOH), and OxPE 18:2_16:1(COOH), alongside a decrease in oxLPE 16:1(CHO).

Due to the impact of COVID-19 infection, an escalation in the levels of oxidized forms of phosphatidylcholines and phosphatidylethanolamines was observed in cord blood. Notably, within all of these forms, oxidation took place in conjunction with palmitoleic acid C16:1. Furthermore, even in the sole oxylipid that exhibited a decrease during the infection, oxLPE 16:1(CHO), C16:1 underwent oxidation as well.

This phenomenon observed in umbilical cord blood could conceivably be linked to the initiation of an oxidative cascade during the transition from the prenatal to postnatal environment, coinciding with the commencement of spontaneous breathing in the newborn. However, considering the statistically significant alterations in the levels of oxidized forms within the primary group, it is reasonable to hypothesize a connection with the processes of oxidative stress associated with viral infection.

When we were composing our article, we did not encounter any existing literature describing this phenomenon in cord blood. This observation has been included in the article to address this gap in knowledge.

There was a pronounced increase in the levels of nearly all marker lipids, particularly lysophospholipids, triglycerides, phosphatidylcholines, sphingomyelins, and oxidized lipids. In addition, levels of cholesterol esters 16:1 and 20:3, phosphatidylcholines 16:0_20:3, MGDG 18:1_22:6, OxPC 16:1_16:1(COOH), and SM d24:1/18:1 increased, and TG 18:0_18:1_18:1 decreased when the COVID-19 infection occurred closer to the delivery date (Appendix A).

COVID-19 infection exerts a diverse impact on the lipid composition of amniotic fluid during delivery (Figure 3, Appendix A). Nine lipids show a decrease, while six lipids show an increase compared to the control group. Specifically, a significant decrease in PG, LPG, and PI was observed during infection in the second and third trimesters of pregnancy. For comprehensive information on lipid markers at individual stages of infection in the three biological fluids, please refer to Appendix A.

The specificity of maternal plasma lipidome concerning the infection history during pregnancy suggests the potential for predicting complications associated with COVID-19, particularly the health status of the child. This will enable the identification of a group of children requiring more vigilant monitoring during the neonatal period. To simplify the notation within the model, we designated children who required stay and therapy in the intensive care unit and/or neonatal pathology department with the term “presence of pathology”, thereby indicating their difference from healthy infants. Healthy children are full-term newborns without somatic and neurological pathologies, discharged home in satisfactory condition on time. Logistic regression models were created for both the control group and patients who experienced COVID-19 in the first to third trimesters of pregnancy to determine the presence of pathology in the newborn based on the maternal plasma lipidome (refer to Table 4 and Appendix A). Each model demonstrates good diagnostic value, with sensitivity of at least 0.75, specificity of at least 0.86, and an area under the ROC curve of at least 0.87.

The predominant complication observed in newborns following COVID-19 infection in the mother during the second trimester of pregnancy manifested as congenital heart defects (CHD). Endeavors were undertaken to develop predictive models for MPS. Through mathematical data processing, these models were constructed to forecast MPS occurrence in patients exposed to COVID-19 during the second trimester, as well as for the control group. Thorough analysis of the variables integrated into the prognostic models for both the primary and control groups facilitated the identification of lipids that exhibited a close association with COVID-19.

For the control group, a predictive model was formulated to ascertain the presence of CHD with a sensitivity of 1 and a specificity of 0.95. This model hinged on lipids including PC 18:2_18:2, OxCL 18:2_22:6_18:3_22:5(OOH)2, PC 18:0_18:2, and PC O-18:0/18:1 in maternal plasma (outlined in Appendix A).

In instances of COVID-19 infection during the second trimester, the model encompassed CE 16:0 × TG 18:1_22:5_8:0, LPC 16:0 × PE 18:0_20:1, PC 16:0_20:3 × TG 18:1_18:1_20:1, and PC 16:0_16:0 × SM d18:1/18:3. This model demonstrated sensitivity and specificity values of 0.79 and 0.82, respectively (specified in Appendix A).

## 3. Discussion

Based on the data obtained, we observed interconnected changes in the relative concentration of several lipids in the mother–placenta–fetus system, indicating the dependence of metabolic processes in both the mother and fetus, and their deviations in COVID-19. Simultaneous analysis of lipidome in three locations (venous and umbilical blood and amniotic fluid), reflecting the mother–placenta–fetus system, revealed significant changes in lipid levels, predominantly showing an increasing trend in individuals with a history of COVID-19 (Appendix A).

During infection, the mother’s blood plasma showed increased relative concentrations of lysophospholipids, phosphatidylcholines, and ethanolamines, as well as triglycerides and phosphatidylcholines, with the increase becoming more pronounced from the first to the third trimester. Additionally, sphingomyelins were selectively increased during infection in the second trimester.

Lysophospholipids serve as precursors of arachidonic acid (AA, 20:4) and act as second intracellular messengers [10]. These lipids are involved in triggering tissue inflammation processes, leading to impaired hemostasis [11]. Szczuko M. et al. conducted a literature review in 2020 [10] that described the role of pro-inflammatory mediators derived from arachidonic acid in pathological conditions related to reproduction and pregnancy. The review presents evidence from various studies demonstrating the significant impact of uncontrolled inflammation on reproduction, spermatogenesis, endometriosis, the development of polycystic ovary syndrome, implantation, pregnancy, and childbirth. The authors also emphasize that excessive inflammation can lead to miscarriage and other pathological complications during pregnancy [10].

One of the key mechanisms behind this may be the systemic inflammatory response triggered by the production of arachidonic acid. Arachidonic (20:4) and linolenic acids (LA, 18:3) are components of cell membrane phospholipids, which are released under the influence of phospholipases. These acids undergo further transformations through the cyclooxygenase or lipoxygenase pathway [12]. Metabolites of arachidonic acid play important regulatory roles [13,14] and include lipid inflammatory mediators such as prostaglandins, thromboxanes, and leukotrienes, which possess vaso- and bronchoactive properties. The platelet-activating factor, a potent spasmogen, is also formed from membrane phospholipids. This group also encompasses lipid peroxidation products, i.e., lipoperoxides. Fatty acids in cells and tissues are part of various lipid classes, contribute to steroid synthesis, and serve as precursors of prostanoids, promoting free oxidation processes, leading to the formation of peroxidation products and prostaglandins [15,16]. The accumulation of arachidonic acid metabolites in the body contributes to an inflammatory component, which is, however, normally present in a pregnant woman’s body. Therefore, it was essential to compare the relative concentration of the lysophospholipids group during pregnancy in different trimesters with COVID-19 infection against the control group (relatively healthy pregnant women who avoided COVID-19 infection during gestation). This observation may indicate the initiation of an abnormal inflammatory cascade during pregnancy, originating from arachidonic acid precursors in the COVID-19 survivor groups.

In cord blood, the most significant changes were observed in relation to the triglyceride level. For 19 lipids in this class, there was a statistically significant increase in the relative concentration of triglycerides depending on the time of COVID-19 infection. The fatty acid composition of these triglycerides is highly diverse, consisting of both essential (α-linolenic, 18:3, and linoleic acid, 18:2) and nonessential fatty acids (myristic acid 14:0, palmitic acid 16:0, palmitoleic 16:1, stearic acid 18:0, oleic acid 18:1, gamma-linolenic acid 18:3, stearidonic 18:4, dihomo-gamma-linolenic acid 20:3, arachidonic acid 20:4, eicosapentaenoic acid 20:5, docosapentaenoic acid 22:5, and docosahexaenoic acid 22:6), with a marked predominance of linoleic (LA, a precursor to AA), palmitic, oleic, and docosahexaenoic (DHA) acids. Triglycerides in the blood are the primary carriers of fatty acids. Triglyceride and cholesterol concentrations increase during pregnancy to meet the growing fetus’s requirements [7]. Pregnant women with low cholesterol have a higher risk of complications, such as fetal growth restriction (FGR) and preterm birth [17]. High concentrations of total cholesterol and triglycerides in the mother are associated with an increased risk of hypertensive disorders, large for gestational age fetus, and, consequently, induced preterm birth [18].

Fatty acids play a crucial role in the structural and functional development of the brain. Therefore, docosahexaenoic acid 22:6 (DHA) is involved in various processes such as angiogenesis, inflammatory response, apoptosis, and cell proliferation, and it is critical for the proper structure and development of the growing fetal brain in the womb [19,20,21,22].

Cord blood shows a similar trend of increasing lysophospholipids, phosphatidylcholines, sphingomyelins, and oxidized lipids, as seen in maternal blood. Additionally, a statistically significant increase in the levels of cholesterol esters of arachidonic and palmitoleic acids (CE 16:1 and CE 20:3) in cord blood was observed, depending on the trimester during which COVID-19 was contracted. Cholesterol fatty acid esters are essential components of the body and crucial for the normal formation of cell membranes. The result of lipid hydrolysis is the formation of glycerol and corresponding carboxylic acids. The tricarboxylic acid cycle is a pivotal stage of respiration in all cells, serving as the central nexus where multiple metabolic pathways intersect within the organism. It underlies viral infection: sustaining replication, viral pathogenesis, and antiviral immunity [23,24,25].

The levels of phosphatidylinositols (PI 16:0_16:0, PI 16:0_18:1, PI 18:0_18:2, PI 18:0_20:3, PI 18:0_20:4) in maternal plasma were reduced during COVID-19 in the first and second trimesters. Notably, all these PIs contain one unsaturated fatty acid (palmitic or stearic). The crucial role of PIs lies in their association with the triggering of a cascade of thrombogenic processes through the formation of antibodies against them. These antibodies, categorized as anionic antiphospholipid antibodies, target negatively charged phospholipids like phosphatidic acid, phosphatidylserine, and phosphatidylinositol [26]. In vitro models have shown that these antibodies can disrupt trophoblast invasion and syncytiotrophoblast formation, leading to reduced hCG levels [27,28]. The decrease in phosphatidylinositols levels in patients after recovering from COVID-19 may be related to their involvement in autoimmune complexes that contribute to increased thrombus formation. Currently, patients with COVID-19 are known to be at high risk of thrombosis [29]. According to the indirect results of our data, this risk may persist for an extended period, even up to delivery.

When analyzing maternal and umbilical cord blood lipidomes, the most significant change in sphingomyelin levels was observed when COVID-19 infection occurred during the second trimester. Sphingomyelins are vital components of cell membranes, particularly the myelin sheath surrounding nerve cell axons. They are also involved in cell signal transduction. Abusukhun M. et al. discovered activation of the sphingomyelinase-ceramide pathway in 23 intensive care patients with severe COVID-19 [30]. The authors noted an increase in circulating sphingomyelinase activity, leading to disruption of sphingolipids in serum lipoproteins and erythrocytes. The results of their study revealed a correlation with biomarkers of severe clinical phenotype and confirmed the significant pathophysiological role of acid sphingomyelinase during SARS-CoV-2 infection.

In amniotic fluid during infection in the second and third trimesters of pregnancy, a statistically significant decrease in PG, LPG, and PI is noteworthy. Phosphatidylglycerol is a glycerophospholipid found in pulmonary surfactant [31] and the plasma membrane, where it directly activates lipid-gated ion channels. The head group substituent glycerol bonded through a phosphomonoester serves as the precursor of surfactant. Its presence in the amniotic fluid of the newborn indicates fetal lung maturity. Together with phosphatidylinositol (PI), PG plays a key role in regulating the innate immune response in the lungs.

An in-depth analysis of clinical and laboratory parallels among the studied groups who underwent COVID-19 during pregnancy (first, second, or third trimesters) revealed statistically significant differences, depending on the time of infection. Regardless of the time of infection, puerperas from the groups who experienced COVID-19 during pregnancy exhibited several disorders in the newborn’s state at the time of delivery, requiring observation of the child under intensive conditions or transfer to the pediatric intensive care unit: with COVID-19 in the first trimester—in 75% (*p* < 0.001), in the second trimester—in 56.7% (*p* = 0.01), and in the third—in 61.2% (*p* = 0.002) of cases. This may be attributed to the implementation of preterm birth (PB) in these groups, leading to the birth of premature newborns who require additional monitoring and therapy. PB occurred in 14.6% of cases in the group that had COVID-19 in the first trimester (*p* = 0.03) and in 11.9% in the second trimester (*p* = 0.03). The frequency of emergency delivery by caesarean section increased significantly more often (18.2%, *p* = 0.01) in the group that experienced COVID-19 in the second trimester. Infection with SARS-CoV-2 in the third trimester did not affect the timing and method of delivery. Only in the group that experienced COVID-19 in the second trimester were maternal and fetal health problems noted, namely the development of GDM that did not require insulin therapy in 19.4% (*p* = 0.01) of cases (vs. 3.4% of GDM in the control group) and the development of CHD in newborns in 33.4% of cases (*p* = 0.02). CHD was also significantly more common in the group of newborns born to mothers who had COVID-19 in the first trimester (42.5%, *p* = 0.003).

Based on a comprehensive analysis of clinical data and plasma lipidome characteristics in patients during delivery, logistic regression models were developed to predict the birth of a child requiring additional observation and/or transfer to the neonatal intensive care unit, as well as the presence of CHD in newborns for patients who underwent COVID-19 and the control group. Notably, the model for the infection group in the second trimester includes not only PC and TG (as in the control and infection groups in the first trimester) but also SM.

In a study conducted by Johannes Kornhuber et al. in 2022, a direct link was established between the course of COVID-19 and the activity of the acid sphingomyelinase enzyme [32]. Acid sphingomyelinase (ASM) cleaves sphingomyelin into a highly lipophilic ceramide, which forms large gel-like rafts/platforms in the plasma membrane. SARS-CoV-2 utilizes these platforms to enter the cell. Inhibition of ceramide by blocking ASM or other methods, such as anticeramide antibodies or degradation by neutral ceramidase, has shown protection against SARS-CoV-2 infection. Clinical trials with drugs that functionally inhibit ASM, known as functional acid sphingomyelinase inhibitors, have demonstrated their potential benefits in COVID-19.

There is a growing body of evidence on the role of sphingolipids in obstetrics. In their work in 2021, Yuliya Fakhr et al. postulate that signaling by the biologically active sphingolipid, sphingosine-1-phosphate (S1P), and its precursors is a novel area of research in pregnancy [33]. S1P and ceramide levels increase towards the end of pregnancy, indicating their physiological role in childbirth. However, elevated levels of circulating S1P and ceramide are associated with pregnancy disorders such as preeclampsia, gestational diabetes mellitus, and intrauterine growth retardation. This aligns with the findings of our study, which showed a statistically significant increase in GDM among pregnant women who experienced COVID-19 in the second trimester (*p* = 0.01).

The expression of placental and decidual enzymes that metabolize S1P and S1P receptors is also impaired in pregnancy complications. S1P signaling mechanisms and ceramide play crucial roles in implantation and early pregnancy by modulating corpus luteum function from progesterone production to luteolysis and apoptosis. Furthermore, S1P plays an important role in inducing decidualization and angiogenesis in the decidua, as well as regulating extravillous trophoblast migration to anchor the placenta in the uterine wall. The study discusses S1P’s role as a negative regulator of trophoblast syncytialization in the multinuclear placental barrier, its role in anti-inflammatory and pro-inflammatory signaling, as well as its function as a vasoconstrictor and the impact of S1P-metabolizing enzymes and receptors during pregnancy [33].

One of the variables in the developed diagnostic model for COVID-19 infection in the third trimester included SM d18:1/24:1 (with a sensitivity and specificity of the model at 0.84 and 0.87, respectively).

Of particular interest is the prognosis of the development of congenital heart defects (CHDs) in children born to mothers who had COVID-19 during pregnancy based on the maternal plasma lipidome. A model with a sensitivity of 0.79, a specificity of 0.82, and a cut-off value of 0.41 was created for a cohort of COVID-19 patients in the second trimester. This model included essential metabolites such as CE 16:0, TG 18:1_22:5_8:0; LPC 16:0, PE 18:0_20:1; PC 16:0_20:3, TG 18:1_18:1_20:1; PC 16:0_16:0, and SM d18:1/18:3. It is evident that this model comprises key metabolites reflecting the crucial pathogenetic processes occurring in the mother–fetus system during a viral infection. These processes include an increase in circulating sphingomyelinase activity, subsequent disruption of sphingolipids in serum lipoproteins and erythrocytes, viral penetration into the body and subsequent spread, and the initiation of inflammatory cascade reactions through the accumulation of arachidonic acid metabolites. The model also demonstrates disruption of fat hydrolysis processes, interfering with the normal functioning of the tricarboxylic acid cycle, and the formation of cell membranes.

The active processes of organogenesis and fine differentiation of fetal organs during the second trimester makes them susceptible to viral damage. This susceptibility is likely reflected in a statistically significant increase in small heart anomalies in fetuses whose mothers experienced COVID-19 in the second trimester. The influence of various viruses on CHD development during pregnancy has been established. Notably, the coxsackie-adenoviral receptor (CAR), a transmembrane cell surface protein responsible for binding Coxsackievirus or adenoviruses, plays a significant role in the pathogenesis of myocarditis. Based on the data obtained, it can be hypothesized that key mechanisms undergoing changes in the mother–fetus system due to COVID-19 involve the accumulation of arachidonic acid metabolites, regulation of the inflammatory response, and possibly activation of the hemostasis system, likely associated with the formation of antibodies to phosphatidylinositols. These processes are most evident during infection in the second trimester, potentially explaining the occurrence of CHDs in newborns. This might be attributed to the timing of organogenesis for the fetus and its significant need for essential building materials in the “fight” against viral aggression, leading to competitive lipid uptake.

## 4. Materials and Methods

### 4.1. Study Design

The study included pregnant women who sought care at the National Medical Research Center for Obstetrics, Gynecology, and Perinatology, named after Academician V.I. Kulakov, under the Ministry of Healthcare of the Russian Federation for pregnancy and childbirth. Inclusion criteria were as follows: signed informed voluntary consent to participate in the study, age between 18 and 45 years, confirmed new coronavirus infection (COVID-19) during pregnancy (main group), and no history of COVID-19 infection during pregnancy (control group).

Exclusion criteria were: refusal to participate in the study, history of COVID-19 vaccination, morbid obesity (BMI ≥ 40.0 kg/m^2^), involvement in donor or surrogacy programs, HIV infection, systemic connective tissue diseases, rheumatic diseases, any type of oncological diseases, severe extragenital pathology in pregnant women, multiple pregnancies, fetal congenital malformations, diabetes mellitus, placenta accreta, and/or placenta previa. Patients who voluntarily decided to withdraw from the study were also excluded.

Data on previous COVID-19 infection were obtained from patient reports, confirmed by information from the Uniform State Health Information System, and/or the presence of a certificate indicating the level of IgG to SARS-CoV-2 in the blood serum above the positivity index (PI).

Samples were collected between October 2019 and January 2021, and the study included 234 pregnant women. Among them, 175 women in the main group had COVID-19 at different stages of pregnancy and were further categorized into three subgroups based on the trimester in which the disease occurred: 41 in the first trimester, 67 in the second trimester, and 67 in the third trimester. The control group consisted of 59 patients who did not have COVID-19 during pregnancy and were not vaccinated (no presence of antibodies to SARS-CoV-2 at the time of delivery). Mass spectrometry analysis was performed on venous blood, cord blood, and amniotic fluid samples from 181 patients.

All patients involved in the study provided voluntary informed consent to participate. The research received approval from the Ethical Committee of the National Medical Research Center for Obstetrics, Gynecology, and Perinatology named after Academician V.I. Kulakov (protocol No. 17 of 23 April 2020).

### 4.2. Sample Collection

Samples of maternal plasma, cord blood plasma, and amniotic fluid were collected at the time of delivery (Appendix A).

Samples of amniotic fluid were collected once during labor and all underwent analysis. The collected amniotic fluid was stored in 15 mL plastic tubes and then subjected to centrifugation (10 min at 12,000 rpm at 40 °C). After centrifugation, the supernatant was transferred into 2 mL cryotubes (1.7–2 mL per tube).

Maternal venous blood was drawn into a 3 mL EDTA tube and subsequently centrifuged at 3000 rpm for 20 min at 40 °C. The resulting supernatant was further centrifuged at room temperature for 10 min at 12,000 rpm and then collected into 2 mL cryotubes (0.5–1 mL per tube).

For newborn cord blood, a 1 mL EDTA tube was used, and the sample preparation procedure was the same as that for maternal blood.

The processing and storage of all samples was conducted within 30 min of extraction. The labeled tubes were then frozen at −80 °C and stored until the time of measurements.

### 4.3. Sample Preparation and Lipidomic Analysis (HPLC-MS/MS)

Lipid extraction from maternal plasma, cord blood, and amniotic fluid followed the modified Folch method [34,35]. For each sample, 40 (500) μL of maternal plasma/cord blood (amniotic fluid) was mixed with 480 (1000) μL of a chloroform–methanol mixture (2/1, *v*/*v*). The mixture was then subjected to ultrasound exposure, and 150 (0) μL of water was added before centrifuging for 10 min at 13,000× *g*. Subsequently, 150 (600) μL of the lower organic layer was carefully transferred to a new vial. The second extraction step involved adding 250 (500) µL of chloroform-methanol mixture (2/1, *v*/*v*) to the remaining sample, followed by ultrasound exposure and another round of centrifugation for 10 min at 13,000× *g*. Next, 300 (400) µL of the lower organic layer was taken and combined with the previously collected 150 (600) µL. The resulting mixture was dried using a nitrogen stream and then redissolved in 200 (100) µL of isopropanol/acetonitrile (1/1, *v*/*v*). After another round of centrifugation for 5 min at 13,000× *g*, 160 (80) µL of the sample was transferred to a vial for subsequent lipid analysis. To prepare quality control (QC) samples, 20 µL from each sample was mixed.

Lipid analysis of the samples was performed using reversed HPLC-MS/MS based on a previously developed protocol [36]. The analysis utilized a Dionex UltiMate 3000 chromatograph (Thermo Scientific, Waltham, MA, USA) coupled to the Maxis Impact qTOF mass spectrometer (Bruker Daltonics, Bremen, Germany) equipped with electrospray ionization. Chromatographic separation is performed by Zorbax C18 column (150 × 2.1 mm, 5 μm, Agilent, Santa Clara, CA, USA) with acetonitrile/water (60:40, *v*/*v*) with of 0.1% formic acid and 10 mM ammonium formate as eluent A and acetonitrile/2-propanol/water, 90:8:2, *v*/*v*/*v*, with 0.1% formic acid and 10 mM ammonium formate as eluent B coupled. The analysis was carried out in both positive and negative ion modes in mass range 100–1700 *m*/*z*, with tandem MS analysis using data-depended analysis mode with 35 eV collision energy, 3 Da isolation window, and mass exclusion time of 1 min. [36]. Data preprocessing and compound identification were conducted according to the protocol developed by J.P. Koelmel et al. using characteristic ion fragments [37] with MzMine software 2.2.6 and Lipid Match 3.5 software by J.P. Koelmel et al. The lipid nomenclature adhered to Lipid Maps. The resulting HPLC-MS/MS data were normalized between batches using autoscaling [38]. Internal standards were not utilized in this study. System performance was monitored through repetitive injections of a quality control sample, yielding lipid signals that exhibited a variance of within 20%. To assess the reproducibility of sample extraction, technical replicates of selected samples were prepared. Additionally, signal normalization was implemented to explore the lipid profile while mitigating potential variations. This methodology has demonstrated favorable outcomes in our prior investigations [38]. The Appendix A provides typical LC–MS data for lipids (Appendix A).

### 4.4. Statistical Analysis

The statistical analysis of clinical data and lipidome data was performed using our custom scripts in the R language (version 4.1.1). Quantitative clinical data and the intensity of lipid peaks were described using median values (Me) and quartiles (Q1, Q3). Qualitative parameters were presented as absolute values and as a percentage of the full clinical group.

To analyze the distribution of qualitative clinical parameters between clinical groups, a Pearson’s chi-squared test was employed. For the comparative analysis of quantitative clinical parameters and the lipidome, the Mann–Whitney test was used for two groups, and the Kruskal–Wallis test was utilized for comparisons involving more than two groups. The significance threshold was set at 0.05.

Models for determining the absence of pathologies in a child and the presence of congenital heart defects (CHDs) for each clinical group (infection in the first trimester, second trimester, and third trimester, and control) were created based on the lipid profile of venous blood. Logistic regression was employed, considering the interaction between variables, and the model took the form
y = 1/(1 + e^(−x × β^T^)),(1)
where the vector x containing independent variables included I_i_, I_i_ × I_j_, and I_i_^2, where I_i_ represents the value of the i-th marker, and β is the vector of coefficients.

New datasets were formed, including lipid levels from both ion regimes, variables representing the result of pairwise multiplication of lipid levels to account for the effect of parameter interactions among themselves, and variables representing the result of squaring lipid levels. From the preliminary set of markers, variables were selected based on their projection value in the analysis of orthogonal projections onto hidden structures, ensuring it exceeded 1. The selection of variables into the model based on logistic regression was carried out step-by-step, using the Akaike information criterion to obtain the minimum value of the criterion at each stage while observing a decrease in the criterion value.

Afterwards, variables were gradually excluded from the model, with coefficients having the highest probability of being equal to zero, until the coefficients for the variables obtained a probability of being equal to zero or less than 0.05.

The quality of the model was assessed using cross-validation on a single object, and sensitivity and specificity were calculated based on its results. The sum of sensitivity and specificity was maximized to evaluate the model’s performance.

## 5. Conclusions

Statistically significant differences between the group of healthy patients and patients who experienced COVID-19 are observed in several lipids in both maternal and cord plasma. These differences manifest as an increase in the level of lysophospholipids, triglycerides, sphingomyelins, and oxidized lipids. Notably, there is a pronounced increase in maternal plasma sphingomyelins and oxidized lipids (CL, LPC, PC, PE) during infection in the second trimester. Moreover, the increase in the content of these lipids in cord blood becomes more significant the less time there is between the COVID-19 infection and delivery.

COVID-19 infection, regardless of the time of infection, exerts a multi-directional effect on the lipid composition of the amniotic fluid obtained during delivery. Nine lipids were reduced, and six were elevated compared to the control group. Specifically, a decrease in the levels of phosphoglycerides, lysophosphoglycerides, and phosphatidylinositols was found during infection in the second and third trimesters of pregnancy. Phosphoglycerides are major components of fetal surfactant and, together with phosphatidylinositols, are key regulators of the innate immune response in the lungs of the newborn.

The specificity of maternal plasma lipidome in relation to the infectious history of pregnancy allows us to propose models for predicting complications associated with COVID-19, particularly in relation to the health of the child. These models can identify a group of children requiring more careful monitoring during the neonatal period. For the control group and patients who experienced COVID-19 in the first to third trimesters of pregnancy, logistic regression models were created to determine the presence of pathology in the newborn based on the maternal plasma lipidome. Each model has a good diagnostic value, with sensitivity above 0.75, specificity above 0.86, and an area under the ROC curve above 0.87.

The main complication observed in newborns after maternal COVID-19 infection during the second trimester of pregnancy was CHDs. A model was created for cases of COVID-19 infection in the second trimester that included lipids (PC 16:0_20:1 × TG 16:0_16:1_18:0, SM d18:0/22:0 × TG 18:1_22:5_8:0, MGDG 16:0_20:0 × SM d18:0/22:0, and PC 18:1_18:3 × OxPC 18:2_16:1(COOH)) and determined the risk of CHDs in the newborn; it has a sensitivity of 0.79 and specificity of 0.82.

In summary, changes in the lipid composition of maternal plasma and umbilical cord blood at delivery associated with COVID-19 infection during pregnancy are synergistic. Notably, the greatest disturbance is associated with infection in the second trimester of pregnancy. Thus, it can be assumed that a severe viral infection affects key metabolic processes in the body of both the mother and the fetus. While we do not observe critical disorders for the mother and fetus under the influence of SARS-CoV-2, the effects we have recorded on the molecular level with COVID-19 should not be ignored, as long-term consequences for the child may manifest themselves at a later age. Therefore, it is essential to monitor the health status of this group of children.

## Figures and Tables

**Figure 1 ijms-24-13787-f001:**
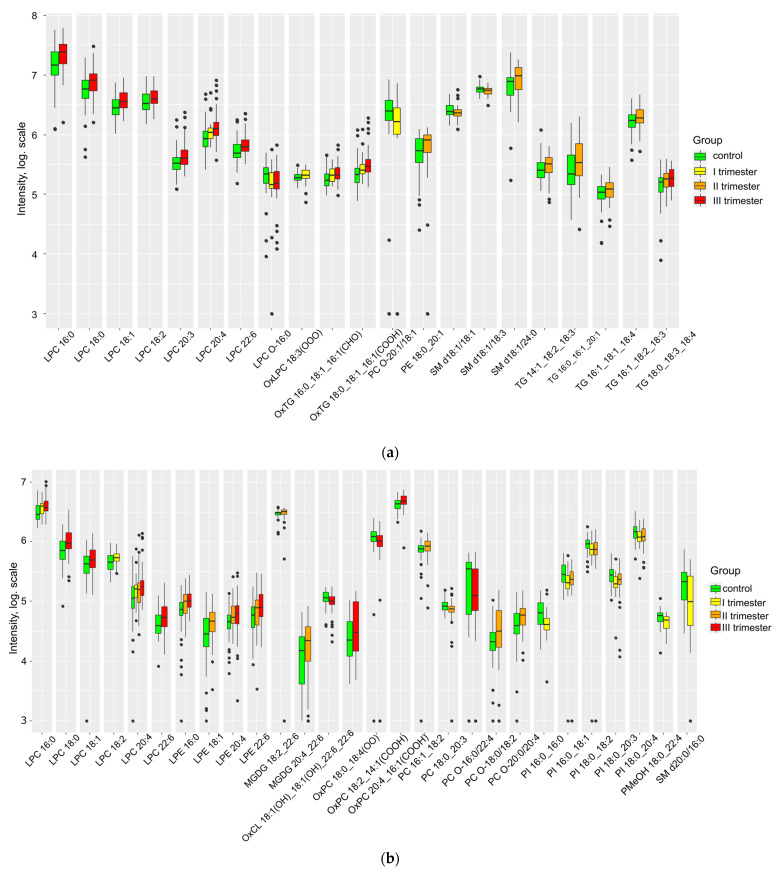
Lipids differing in maternal plasma relative to control in (**a**) positive ion mode and (**b**) negative ion mode.

**Figure 2 ijms-24-13787-f002:**
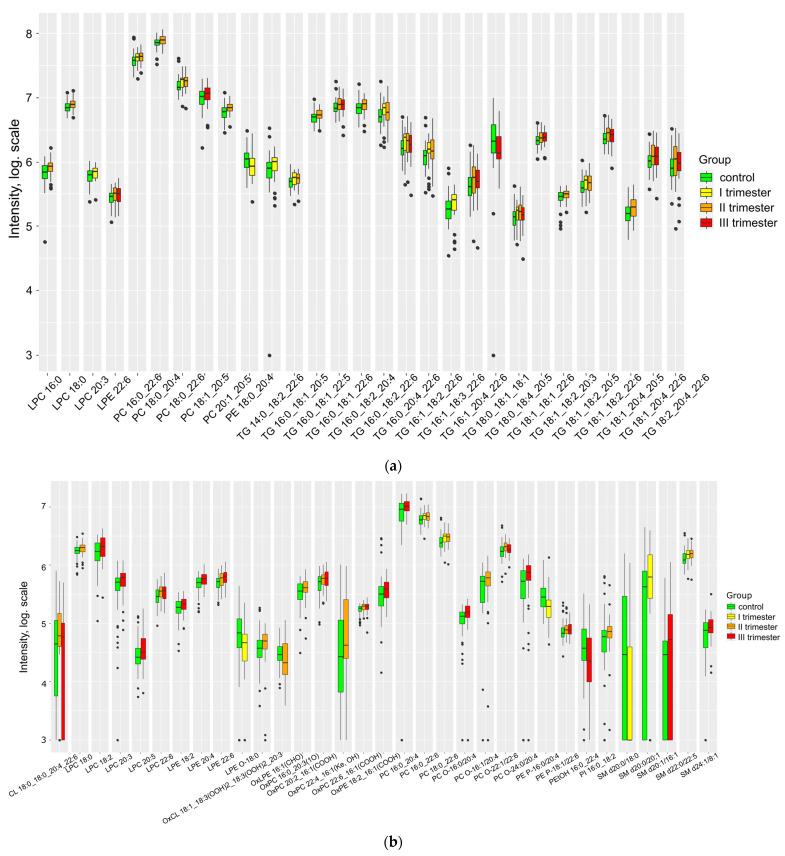
Lipids differing in umbilical cord plasma relative to control (**a**) positive ion mode and (**b**) negative ion mode.

**Figure 3 ijms-24-13787-f003:**
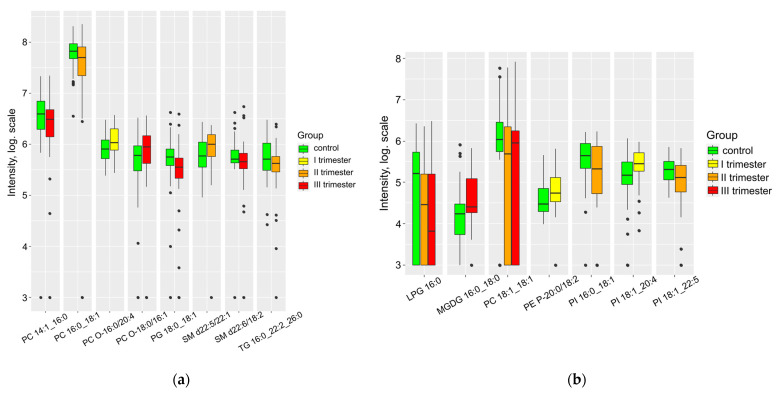
Lipids differing in amniotic fluid relative to control (**a**) positive ion mode and (**b**) negative ion mode.

**Table 1 ijms-24-13787-t001:** Clinical data of pregnant women in three subgroups and the control group.

	Group Ia (*n* = 41)	Group Ib (*n* = 67)	Group Ic (*n* = 67)	Control (*n* = 59)	*p*
Anthropometric data and age					
Age, years	32.0 (28.0;36.0)	32.0 (28.0;34.0)	31.0 (28.3;25.0)	33.0 (29.0;36.0)	0.54
Height, cm	165 (160;171)	165 (162;170)	165 (163;170)	168 (163;170)	0.50
Weight, kg	74.0 (66.3;82.0)	72.0 (67.0;83.5)	73.0 (66.5;78.3)	72.0 (66.0;81.0)	0.88
BMI, kg/m^2^	27.0 (25.0;30.5)	26.0 (24.0;29.0)	27.0 (24.3;29.0)	26.0 (24.0;29.0)	0.66
Lipid metabolism disorders, *n* (%)	26 (63.4)	39 (58.2)	39 (58.2)	34 (57.6)	0.94
Reproductive history					
Number of pregnancies	2.0 (1.0;3.0)	2.0 (1.0;3.0)	2.0 (1.0;3.0)	2.0 (1.0;2.5)	0.95
Medical abortion, *n* (%)	4 (9.8)	10(14.9)	12(17.9)	4 (6.8)	0.25
Non-developing pregnancy, *n* (%)	6 (14.6)	14 (20.9)	16 (23.9)	6 (10.2)	0.19
The course of the first trimester of pregnancy					
Chorion previa *n* (%)	0 (0.0)	1 (1.5)	2 (3.0)	0 (0.0)	0.41
Toxicosis *n* (%)	6 (14.6)	18 (26.9)	17 (25.3)	10 (16.9)	0.32
Anemia before 12 weeks *n* (%)	4 (9.8)	0 (0.0)	0 (0.0)	2 (3.4)	0.007
Exacerbation of chronic diseases before 12 weeks *n* (%)	0 (0.0)	2 (3.0)	1 (1.5)	2 (3.4)	0.64
Hormonal support up to 12 weeks *n* (%)	9 (20.0)	10 (14.9)	16 (23.9)	6 (10.1)	0.18
Gestational diabetes up to 12 weeks *n* (%)	2 (4.9)	1 (1.5)	1 (1.5)	1 (1.7)	0.62
Antibacterial therapy up to 12 weeks *n* (%)	5 (12.1)	3 (4.5)	3 (4.5)	2 (3.4)	0.22
Threatened miscarriage before 12 weeks *n* (%)	8 (19.5)	12 (17.9)	14 (20.9)	6 (10.9)	0.41
The course of the second trimester of pregnancy					
Isthmic-cervical insufficiency *n* (%)	3 (7.3)	4 (6.0)	4 (6.0)	0 (0.0)	0.26
Surgical correction *n* (%)	2 (4.9)	3 (4.5)	2 (3.1)	0 (0.0)	0.41
Obstetric pessary *n* (%)	1 (2.4)	1 (1.5)	3 (4.5)	0 (0.0)	0.37
Threatened miscarriage or preterm birth from 13 to 27 weeks *n* (%)	2 (4.9)	7 (10.4)	7 (10.4)	1 (1.7)	0.17
Anemia from 13 to 27 weeks *n* (%)	4 (9.7)	9 (13.4)	6 (9.0)	8 (13.6)	0.79
Antibacterial therapy from 13 to 27 weeks *n* (%)	3 (7.3)	13 (19.4)	2 (3.0)	0 (0.0)	<0.001
Gestational diabetes from 13 to 27 weeks *n* (%)	3 (7.3)	7 (10.4)	5 (7.5)	1 (1.7)	0.27
The course of the third trimester of pregnancy					
Risk of preterm birth after 28 weeks *n* (%)	5 (12.2)	3 (4.5)	5 (7.5)	4 (6.7)	0.53
Anemia after 28 weeks *n* (%)	12 (29.3)	17 (25.8)	13 (19.4)	21 (35.6)	0.23
Edema of pregnancy *n* (%)	1 (2.1)	5 (7.8)	2 (3.0)	1 (1.7)	0.31
Antibacterial therapy after 28 weeks *n* (%)	0 (0.0)	0 (0.0)	6 (9.0)	0 (0.0)	0.002
Gestational diabetes mellitus after 28 weeks *n* (%)	1 (2.1)	4 (6.1)	0 (0.0)	1 (1.7)	0.16
Terms and methods of delivery					
Delivery time, weeks	39.2 (38.3;40.1)	39.6 (38.4;40.4)	39.4 (38.5;40.1)	39.5 (39.0;40.2)	0.26
Type of delivery *n* (%) Natural birth	29 (70.7)	50 (74.6)	44 (65.7)	39 (66.1)	0.19
Type of delivery *n* (%) Cesarean section	12 (29.3)	17 (25.4)	23 (34.3)	20 (33.9)	
Planned caesarean section, *n* (%)	5 (41.7)	5 (29.4)	10 (50.0)	15 (75.0)	0.04
Emergency caesarean section, *n* (%)	7 (58.3)	12 (70.6)	10 (50.0)	5 (25.0)	
Newborn Data					
Weight, kg	3.4 (2.9;3.6)	3.4 (3.0;3.8)	3.4 (3.1;3.8)	3.5 (3.2;3.7)	0.50
Length cm	53 (51;54)	52 (50;54)	52 (51;54)	53 (51;54)	0.55
Presence of deviations *n* (%)	30 (73.2)	38 (56.7)	41 (61.1)	19 (32.2)	<0.001
Apgar-1 *n* (%) 8	37 (90.2)	60 (89.6)	63 (94.0)	54 (91.5)	0.52
7	4 (9.7)	3 (4.5)	4 (6.0)	4 (6.8)	
6	0 (0.0)	2 (3.0)	0 (0.0)	1 (1.7)	
5	0 (0.0)	2 (3.0)	0 (0.0)	0 (0.0)	
Apgar-5 *n* (%) 9	26 (63.4)	53 (79.1)	56 (83.6)	53 (89.8)	0.007
8	14 (34.1)	9 (13.4)	11 (16.4)	5 (8.5)	
7	1 (2.4)	5 (7.5)	0 (0.0)	1 (1.7)	
Additional chord of the ventricle, *n* (%)	3 (7.3)	13 (19.4)	4 (5.9)	0 (0.0)	<0.001
Interatrial communication *n* (%)	17 (41.5)	22 (32.8)	25 (37.3)	8 (13.6)	0.007

**Table 2 ijms-24-13787-t002:** The course of COVID-19 in 175 patients included in the study.

			Group Ia (*n* = 41)		Group Ib (*n* = 67)		Group Ic (*n* = 67)		*p*	
Treatment	Outpatient, *n* (%)		36 (87.8)		54 (80.6)		56 (83.6)		0.60	
	Hospital, *n* (%)		5 (12.2)		13 (19.4)		11 (16.4)			
The course of the disease	Asymptomatic, *n* (%)		11 (26.8)		20 (29.9)		18(26.9)		0.86	0.55
	Symptoms	temperature, *n* (%)	30 (73.2)	22 (53.7)	47 (70.1)	38 (56.7)	49 (73.1)	33 (49.3)		
		cough, *n* (%)		14 (34.1)		20 (29.9)		23 (34.3)		0.94
		anosmia, *n* (%)		14 (34.1)		32 (47.8)		28 (41.8)		0.11
		pneumonia, *n* (%)		3 (7.3)		11 (16.4)		4 (6.0)		0.06

**Table 3 ijms-24-13787-t003:** Clinical data that statistically significantly change with COVID-19 infection in different trimesters of pregnancy.

Parameter	Control (*n* = 59)	Group I (*n* = 175)	*p*
First trimester of pregnancy			
Delivery time, *n* (%)	Timely—54 (91.5%)	Timely—34 (82.9%)	0.03
	Premature—1 (1.7%)	Premature—6 (14.6%)	
	Belated—4 (6.8%)	Belated—1 (2.4%)	
Apgar-5 min, *n* (%)	9—53 (89.8%)	9—26 (65.0%)	0.009
	8—5 (8.4%)	8—13 (33.5%)	
	7—1 (1.7%)	7—1 (2.5%)	
Baby is healthy, *n* (%)	No—19 (32.2%)	No—30 (75.0%)	<0.001
	Yes—40 (67.7%)	Yes—10 (30.0%)	
Second trimester of pregnancy			
Delivery time, *n* (%)	Timely—54 (91.5%)	Timely—58 (86.6%)	0.03
	Premature—1 (1.7%)	Premature—8 (11.9%)	
	Belated—4 (6.8%)	Belated—1 (1.5%)	
Type of delivery, *n* (%)	Emergency caesarean section—5 (8.6%)	Emergency caesarean section—12 (18.2%)	0.01
	Natural childbirth—38 (64.4%)	Natural childbirth—49 (72.0%)	
	Planned caesarean section—15 (25.4%)	Planned caesarean section—5 (6.5%)	
Gestational diabetes mellitus, *n* (%)	No—56 (94.9%)	No—54 (80.6%)	0.01
	Yes. diet—2 (3.4%)	Yes. diet—13 (19.4%)	
	Yes. insulin—1 (1.7%)	Yes. insulin—0 (0.0%)	
Baby is healthy, *n* (%)	No—19 (32.2%)	No—38 (56.7%)	0.01
	Yes—40 (67.8%)	Yes—29 (43.3%)	
Additional chord, *n* (%)	Yes—0 (0.0%)	Yes—13 (19.7%)	<0.001
	No—59 (100%)	No—53 (80.3%)	
CHDs (Congenital Heart Defects), *n* (%)	No—51 (86.4%)	No—44 (66.6%)	0.02
	Yes—8 (13.4%)	Yes—22 (33.4%)	
Third trimester of pregnancy			
Baby is healthy, *n* (%)	No—19 (32.2%)	No—41 (61.2%)	0.002
	Yes—40 (67.8%)	Yes—26 (38.8%)	

**Table 4 ijms-24-13787-t004:** Diagnostic performance (AUC, sensitivity, specificity) and variables of the models developed.

Model	Sensitivity	Specificity	Cutoff Threshold	AUC ROC	Variables
Control	0.94	0.89	0.63	0.93	PC 16:0_16:0 × TG 14:0_16:1_18:2, OxLPC 18:4(OOO) × PC 16:0_18:2, PC P-16:0/18:2 × TG 16:0_16:0_18:1, PC 16:0_20:5 × OxPC 18:2_16:1(COOH)
COVID-19 in the first trimester	0.89	0.86	0.31	0.87	PC 16:1_18:0 × LPC 16:1
COVID-19 in the second trimester	0.75	0.92	0.37	0.88	PC 16:0_20:1 × TG 16:0_16:1_18:0, SM d18:0/22:0 × TG 18:1_22:5_8:0, MGDG 16:0_20:0 × SM d18:0/22:0, PC 18:1_18:3 × OxPC 18:2_16:1(COOH)
COVID-19 in the third trimester	0.88	0.89	0.28	0.89	SM d18:1/24:1 × PC 18:1_20:1, PC 16:0_22:6 × PC O-16:0/20:4, PC 16:1_22:6 × PC 16:0_22:6, LPC 16:0 × PC P-20:1/18:1

## Data Availability

Data are contained within the Appendix A.

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
