# Peer review of "COVID-19 Infection during Pregnancy: Disruptions in Lipid Metabolism and Implications for Newborn Health"

_ijms, 2023, doi:10.3390/ijms241813787_

Round 1

Reviewer 1 Report

The authors analyzed lipid profiles in pregnant women and connected the issue of newborn development concerning COVID-19. The topic was meaningful, the data amount was large, and the findings sounded reasonable.

1. Line 262: What was the predominant PI species in plasma samples in this study? Usually, it should be PI 18:0/20:4. Therefore, I am afraid that the species listed here were actually minor ones, of which the levels were not reliable. Yet, I would like to give some comments on the aspect of lipidomic analysis in this work.

2. The authors thought of COVID-19 infection as somehow related to inflammation. So, they may consider analyzing the fatty acyl composition within the same lipid class between different groups.

3. I strongly suggest the authors show their results using chemometric tools. With so many variables, it would be hard for readers to quickly catch the points.

4. Supplementary 3 is not so meaningful in my opinion. TIC shows nothing informative in this lipidomic study. Same issue for Supplementary 4.

5. Ionization for CE using ESI is always not so reliable, especially in (semi)-quantitative studies.

6. Although oxidized lipids (CL, LPC, PC, PE) were identified based on LIPIDMAPS, it would be better to set up an in-house library for more reliable identification, since oxidized lipids are always a mixture of isomers.

7. Line 162: The diagnostic performance of OxLPC 18:4(OOO) and OxPC 18:2_16:1(COOH) looked nice, but how about other oxLPC and oxPC and other ox-phospholipid species? Please the authors explain the results objectively.

English writing should be improved. There were some sentences that were difficult to read. The authors may consider expressing the ideas in a more concise, more straightforward way. Especially, in the Discussion part, it would be more important to disclose new findings and summarize the results, rather than explain the basic information.

Author Response

Line 262: What was the predominant PI species in plasma samples in this study? Usually, it should be PI 18:0/20:4. Therefore, I am afraid that the species listed here were actually minor ones, of which the levels were not reliable. Yet, I would like to give some comments on the aspect of lipidomic analysis in this work.

Answer:  A typographical error crept into the article, and it is indeed PI 18:0_20:4, as reflected in Figure 7. Furthermore, it can be observed from the figure that PI 18:0_20:4 predominates over other PIs, in accordance with your statement

  1. The authors thought of COVID-19 infection as somehow related to inflammation. So, they may consider analyzing the fatty acyl composition within the same lipid class between different groups.

Answer: The majority of changes in the lipid profile in this study are associated with lipid classes (meaning that typically, changes within one lipid class occur in the same direction). The emerging differences, which can be associated with changes in the fatty acid composition, are not sufficient to support any claims.

  1. I strongly suggest the authors show their results using chemometric tools. With so many variables, it would be hard for readers to quickly catch the points.

Answer: We added in supplementary figures (Supplementary S2, Figure 1S, 3S, 5S), which are presented samples in PCA-space.

  1. Supplementary 3 is not so meaningful in my opinion. TIC shows nothing informative in this lipidomic study. Same issue for Supplementary 4.

Answer: Sometimes, during publication, there is a request to attach chromatograms and mass spectra of samples. Unfortunately, the differences in the lipid profile in this task are not significant enough to be represented by an averaged mass spectrum. Therefore, Appendices 3 and 4 (abbreviated) are mainly of illustrative nature.

  1. Ionization for CE using ESI is always not so reliable, especially in (semi)-quantitative studies.

Answer:  We partially agree with this remark. Although in several studies such as "Shotgun lipidomics on a LTQ Orbitrap mass spectrometer by successive switching between acquisition polarity modes" by Schuhmann K. et al, "A novel informatics concept for high-throughput shotgun lipidomics based on the molecular fragmentation query language" by Herzog et al, and "Molecular lipid species in urinary exosomes as potential prostate cancer biomarkers" by Skotland et al, cholesterol esters are detected in positive mode along with lipids of other classes, which is what we also did.

  1. Although oxidized lipids (CL, LPC, PC, PE) were identified based on LIPIDMAPS, it would be better to set up an in-house library for more reliable identification, since oxidized lipids are always a mixture of isomers.

Answer: Thank you for the remark, we will strive to improve our future research according to your recommendations.

  1. Line 162: The diagnostic performance of OxLPC 18:4(OOO) and OxPC 18:2_16:1(COOH) looked nice, but how about other oxLPC and oxPC and other ox-phospholipid species? Please the authors explain the results objectively.

The text was added:

A total of 2 OxLPC and 25 OxPC were identified. Of these lipids, only OxLPC 18:4(OOO) and OxPC 18:2_16:1(COOH) showed biomarker potential. In the maternal plasma of the primary group, a noticeable elevation was observed in the concentrations of OxLPC18:3(OOO), OxTG 16:0_18:1_16:1(CHO), OxTG 18:0_18:1_16:1(COOH), OxPC 18:0_18:4(OO), OxPC 20:4_16:1(COOH), alongside a reduction in OxCL 18:1(OH)_18:1(OH)_22:6_22:6 and OxPC 18:2_14:1(COOH). A broad spectrum of lipids that underwent statistically significant changes during the Covid-19 period (LPC, TG, PC, CL) were found to have undergone oxidation. Notably, oleic acid C18:1 and palmitoleic acid C16:1 were the primary substrates of oxidation.

Within the umbilical cord plasma of the primary group, there was a notable rise in OxCL 18:1_18:3(OOH)2_18:3(OOH)2_20:3, OxPC 16:0_20:3(1O), OxPC 20:2_16:1(COOH), OxPC 22:4_16:1(Ke, OH), OxPC 22:6_16:1(COOH), OxPE 18:2_16:1(COOH), coupled with a decrease in oxLPE 16:1(CHO). The impact of Covid-19 infection led to an increase in the levels of oxidized forms of phosphatidylcholines and phosphatidylethanolamines within umbilical blood, with oxidation primarily driven by palmitoleic acid C16:1. Remarkably, even in the solitary oxylipid that demonstrated reduction during the infection, oxLPE 16:1(CHO), C16:1 underwent oxidation.

In the context of umbilical cord blood, this observation can conceivably be linked to the initiation of an oxidative cascade during the shift from the prenatal to postnatal environment, coinciding with the commencement of spontaneous breathing in the newborn. Nonetheless, considering the statistically significant alterations in the levels of oxidized forms within the primary group, it is reasonable to hypothesize a connection with oxidative stress processes associated with viral infection. At the time of composing our article, we did not come across any existing literature describing this phenomenon in cord blood. This addition has been incorporated into the article to address this gap.

English writing should be improved. There were some sentences that were difficult to read. The authors may consider expressing the ideas in a more concise, more straightforward way. Especially, in the Discussion part, it would be more important to disclose new findings and summarize the results, rather than explain the basic information.

Answer:

We checked the English and rewrote the "Discussion" part.

Reviewer 2 Report

In the study, Frankevich and co-authors investigated the characteristics of the lipid profile in the mother-placenta-fetus system among women who had experienced SARS-CoV-2 infection during pregnancy. The lipidomics analysis was carried out using  HPLC-MS/MS. 

However, prior to publication, substantial revisions are required, with particular emphasis on methodological sections, particularly in the domain of data analysis:

A) Within the "4.3. Sample Preparation and Lipidomic Analysis (HPLC-MS/MS)" section, a more comprehensive explanation of the HPLC-MS/MS protocols must be reported, even though the protocol has already been published (doi:10.1038/s41598-021-89859-0) details about the HPLC conditions and MS data acquisition must be briefly described. Moreover, considering that the method is semi-quantitative, as stated by the authors, the specifics of the internal standards mix, including their concentration and if they are deuterated, employed for relative lipid class quantification must be disclosed. This information is essential to facilitate the reproducibility crisis of the experiment.

B) The "4.4. Statistical Analysis" segment leaves ambiguity regarding the multivariate statistical analysis that preceded the logistic regression. Lines 490-494 hint at the utilization of OPLS-DA, yet no details about the analysis's characteristic values  (e.g., VIP) or graphical representations (e.g. Score plot) are furnished. Supplementary tables present solely the p-values of the most significant lipids. The authors must clarify this aspect. They must report the OPLS-DA score plots in a Figure and the VIP values of the selected biomarker lipids for the logistic regression in a Table. Moreover, they should report the PLS-DA, R2 and Q2 quality values in cross-validation. 

Finally, a minor note, in the Results section, lines explicitly 80-82 must be removed. They are part of MDPI's template instructions, unfitting within the scientific narrative.

Please double-check the English; the manuscript is hard to read in some parts.

Author Response

A) Within the "4.3. Sample Preparation and Lipidomic Analysis (HPLC-MS/MS)" section, a more comprehensive explanation of the HPLC-MS/MS protocols must be reported, even though the protocol has already been published (doi:10.1038/s41598-021-89859-0) details about the HPLC conditions and MS data acquisition must be briefly described. Moreover, considering that the method is semi-quantitative, as stated by the authors, the specifics of the internal standards mix, including their concentration and if they are deuterated, employed for relative lipid class quantification must be disclosed. This information is essential to facilitate the reproducibility crisis of the experiment.

Answer: Description of HPLC-MS analysis were expanded. The internal standards were not used in this study. The performance of the system was controlled by repetitive injection of a quality control sample. The signal of the lipids varied within 20%. The reproducibility of sample extraction was checked by preparing technical replicas of some samples. Besides, signal normalization was used to study lipid profile and reduce possible variations. This approach has shown good results in our previous studies Tokareva, A.O.; Chagovets, V. V.; Kononikhin, A.S.; et al «Normalization methods for reducing interbatch effect without quality control samples in liquid chromatography-mass spectrometry-based studies.» Anal. Bioanal. Chem.(2021), Tokareva, A.O.; Chagovets, V. V.; Kononikhin, A.S.; et al «Pipeline of mass-spectrometry data processing for diagnostic molecular marker panel obtaining using the example of search markers of breast cancer metastasis» Biomedical Chemistry: research and methods (2021)

B) The "4.4. Statistical Analysis" segment leaves ambiguity regarding the multivariate statistical analysis that preceded the logistic regression. Lines 490-494 hint at the utilization of OPLS-DA, yet no details about the analysis's characteristic values (e.g., VIP) or graphical representations (e.g. Score plot) are furnished. Supplementary tables present solely the p-values of the most significant lipids. The authors must clarify this aspect. They must report the OPLS-DA score plots in a Figure and the VIP values of the selected biomarker lipids for the logistic regression in a Table. Moreover, they should report the PLS-DA, R2 and Q2 quality values in cross-validation.

Answer: VIP’s values were added in supplementary tables (table 4S -  and new supplementary table with OPLS-DA model quality characteristic was added in supplementary 2 (table 3S).

Please double-check the English; the manuscript is hard to read in some parts.

Answer:

The English was checked and some parts of the manuscript were rewritten.

Reviewer 3 Report

The manuscript by Frankevich et al describe the changes in mother-placenta-fetus system lipidome caused by the recent COVID-19 epidemics. The work is a valuable contribution to our understanding of the influence of viral infection suffered at different trimesters on the outcomes of pregnancy and potential future health outcomes of the children.

However, the authors should take into consideration the following comments in order to improve the quality of the manuscript:

Line 20: Mann-Whitney

Lines 80-82: remove

Table 1, Reproductive history (abortion and non-developing pregnancy): the numbers in parenthesis represent percentages? The same for the values in Table 2.

Line 430-431: Were the amniotic fluid samples collected at 2 different time points? If so, were they all analyzed?

Line 113: Please, define MPS and RA (line 312) the first time they are mentioned.

Is there any particular reason for not including control group values in Fig S2 and S3 (Supplement2)? Also, it is not clear why some metabolites were presented in the Supplement2 figures and not the main text?

Lines 165-169 claim that the model to determine the development of CHD in the control group was developed. However, in Supplement2, Table 6S, the model variables for healthy babies in the 3rd trimester COVID are presented. Additionally, Tables 7S and 8S present variables for MPS in second trimester and controls, which were not commented on in the main text. Model variables for CHD are not presented at all. Methods section mentions that the logistic regression was used to create models for absence of pathologies and CHD. Please, adjust this section with the Results. Also, to what type of pathology is Table 3 referring to? Also, which parameters were used to define a healthy baby?

In Tables 1-3, some p values seem to be missing.

Please, check the percentage of controls with lipid disorders (Table 1).

Author Response

Line 20: Mann-Whitney

Answer: Corrected.

Lines 80-82: remove

Answer: Corrected.

Table 1, Reproductive history (abortion and non-developing pregnancy): the numbers in parenthesis represent percentages? The same for the values in Table 2.

Answer: Yes, we are added explanations in the Tables 1-3.

Line 430-431: Were the amniotic fluid samples collected at 2 different time points? If so, were they all analyzed?

The text was added: “Samples of amniotic fluid were collected once during labor and all underwent analysis”.

Line 113: Please, define MPS and RA (line 312) the first time they are mentioned.

Answer: We apologize, it was a formatting and translation error in the text., It must be “CHD” and it has been corrected. AR and PR is mistype of PB – “preterm birth”. It has been corrected.

Is there any particular reason for not including control group values in Fig S2 and S3 (Supplement2)? Also, it is not clear why some metabolites were presented in the Supplement2 figures and not the main text?

Answer: Control group was not included in their comparison, in figure S2 and S3 we demonstrated differences in covid-infection group between trimester subgroups. We added come comment about lipid levels in cord in case of various trimester (lines 136-142).

Lines 165-169 claim that the model to determine the development of CHD in the control group was developed. However, in Supplement2, Table 6S, the model variables for healthy babies in the 3rd trimester COVID are presented. Additionally, Tables 7S and 8S present variables for MPS in second trimester and controls, which were not commented on in the main text. Model variables for CHD are not presented at all. Methods section mentions that the logistic regression was used to create models for absence of pathologies and CHD. Please, adjust this section with the Results. Also, to what type of pathology is Table 3 referring to? Also, which parameters were used to define a healthy baby?

Answer: Sorry, “MPS” is artefact of translation, it must be “CHD” and it has been corrected.

Initially, the article presents a comprehensive depiction of the derived models aimed at predicting the likelihood of childbirth requiring intensive observation, transfer to the intensive care unit, or progression to the secondary stage of care, within the neonatal pathology department. These models were formulated for individuals who encountered Covid-19 during the first, second, and third trimesters of pregnancy, as well as for the control group consisting of those unaffected by Covid-19. The corresponding data is outlined in Table 3. To differentiate these newborns from healthy counterparts, we coined the term "presence of pathology" to denote infants necessitating care and treatment in the intensive care unit or neonatal pathology unit.

Furthermore, we have supplemented the text with a clarification: "Healthy children denote infants born at full term without somatic or neurological abnormalities, who were discharged in satisfactory condition within the stipulated timeframe." This elucidation serves to inform readers about the designations within the model and the classification of children into specific groups. This supplementary information has been seamlessly incorporated into the manuscript.

Subsequently, we delve into the description of predictive models tailored to anticipate the emergence of minor cardiac anomalies in newborns. These models were established for both the control group and specifically for patients who recuperated from Covid-19 during the second trimester. For the sake of brevity, we present comprehensive tabular data for these two models in the Appendix, so as not to unduly augment the manuscript's length.

The article proceeds with a discussion on "CHD" as the principal complication found in newborns following maternal Covid-19 infection during the second trimester of pregnancy. Endeavors were undertaken to formulate predictive models for CHD. The development of these models involved mathematical analysis of the data, leading to the creation of predictive models for CHD occurrence among patients exposed to Covid-19 during the second trimester and the control group. Scrutiny of variables integrated into the prognostic models, both for the primary and control groups, enabled the identification of lipids intricately associated with Covid-19.

For the control group, a model was crafted to ascertain the presence of CHD with sensitivity at 1 and specificity at 0.95. This was based on lipids including PC 18:2_18:2, OxCL 18:2_22:6_18:3_22:5(OOH)2, PC 18:0_18:2, and PC O-18:0/18:1 in maternal plasma (detailed in Supplementary S2, Table 3S, Table 8S).

When dealing with Covid-19 infection in the second trimester, the model incorporated CE 16:0 * TG 18:1_22:5_8:0, LPC 16:0 * PE 18:0_20:1, PC 16:0_20:3 * TG 18:1_18:1_20:1, and PC 16:0_16:0 * SM d18:1/18:3. The corresponding model exhibited sensitivity and specificity of 0.79 and 0.82, respectively (outlined in Supplementary S2, Table 3S, Table 9S).                                          

In Tables 1-3, some p values seem to be missing.

Answer: You might not have seen the P-value due to incorrect table formatting. We rechecked them. The style of the tables may not accurately represent categorical parameters, and P-values might appear to be missing.

Please, check the percentage of controls with lipid disorders (Table 1).

Answer: It was corrected, thank you.

Round 2

Reviewer 1 Report

The authors did proper modifications and improved the quality of this manuscript.

Reviewer 2 Report

the manuscirpt could be published in the present form